# Multiscale Structure of Starches Grafted with Hydrophobic Groups: A New Analytical Strategy

**DOI:** 10.3390/molecules25122827

**Published:** 2020-06-18

**Authors:** Chloé Volant, Alexandre Gilet, Fatima Beddiaf, Marion Collinet-Fressancourt, Xavier Falourd, Nicolas Descamps, Vincent Wiatz, Hervé Bricout, Sébastien Tilloy, Eric Monflier, Claude Quettier, Ahmed Mazzah, Agnès Rolland-Sabaté

**Affiliations:** 1University Lille, CNRS, USR3290—MSAP—Miniaturisation pour la Synthèse, l’Analyse et la Protéomique, F-59000 Lille, France; chloe.volant@univ-ubs.fr (C.V.); ahmed.mazzah@univ-lille.fr (A.M.); 2University Artois, CNRS, Centrale Lille, University Lille, UMR 8181—UCCS—Unité de Catalyse et Chimie du Solide, F-62300 Lens, France; alexandre.r.gilet@gmail.com (A.G.); herve.bricout@univ-artois.fr (H.B.); sebastien.tilloy@univ-artois.fr (S.T.); 3INRAE, UR BIA, F-44316 Nantes, France; fb.beddiaf@gmail.com (F.B.); xavier.falourd@inrae.fr (X.F.); 4CIRAD, UPR Recyclage et Risque, F-97743 Saint-Denis, Réunion, France; marion.collinet@cirad.fr; 5University Montpellier, Recyclage et Risque, CIRAD, 34398 Montpellier, France; 6INRAE, BIBS Facility, F-44316 Nantes, France; 7ROQUETTE Frères, Rue de la Haute Loge, 62136 Lestrem, France; nicolas.descamps@roquette.com (N.D.); vincent.wiatz@roquette.com (V.W.); claude.quettier@roquette.com (C.Q.); 8INRAE, Université d’Avignon, UMR SQPOV, F-84914 Avignon, France

**Keywords:** acetylated starch, etherified starch, chemical composition, macromolecular characteristics, surface characterization

## Abstract

Starch, an abundant and low-cost plant-based glucopolymer, has great potential to replace carbon-based polymers in various materials. In order to optimize its functional properties for bioplastics applications chemical groups need to be introduced on the free hydroxyl groups in a controlled manner, so an understanding of the resulting structure-properties relationships is therefore essential. The purpose of this work was to study the multiscale structure of highly-acetylated (degree of substitution, 0.4 < DS ≤ 3) and etherified starches by using an original combination of experimental strategies and methodologies. The molecular structure and substituents repartition were investigated by developing new sample preparation strategies for specific analysis including Asymmetrical Flow Field Flow Fractionation associated with Multiangle Laser Light Scattering, Nuclear Magnetic Resonance (NMR), Raman and Time of Flight Secondary Ion Mass spectroscopies. Molar mass decrease and specific ways of chain breakage due to modification were pointed out and are correlated to the amylose content. The amorphous structuration was revealed by solid-state NMR. This original broad analytical approach allowed for the first time a large characterization of highly-acetylated starches insoluble in aqueous solvents. This strategy, then applied to characterize etherified starches, opens the way to correlate the structure to the properties of such insoluble starch-based materials.

## 1. Introduction

Starch is, after cellulose and hemicelluloses, one of the most abundant carbohydrates in plants. Its semi-crystalline granules are mainly composed of two glucopolysaccharides: amylose, a quasi-linear chain of d-glucosyl units linked in α-1,4, and amylopectin which is constituted by a complex arborescent arrangement of linear chains of d-glucosyl units linked in α-1,4 with about 5–6% of branching points in α-1,6 [1]. Starch is abundant, biodegradable and low cost, and it can easily be processed by extrusion or thermomoulding to obtain biomaterials [2,3]. Unfortunately starch hydrophilicity and poor mechanical properties make it not suitable for bioplastic application [3]. A way to improve its physicochemical properties is introducing functional groups on the free hydroxyl groups by esterification or etherification with a maximum theoretical degree of substitution (DS) of 3 [4]. The properties of such chemically modified starches depend on the starch structure, the DS, the type of grafted groups and their repartition on the macromolecules [5] but these relations are not well known because of structure analysis limitations.

Indeed, because of the physicochemical impacts of starch modifications, the characterization of low DS and high DS modified starch-based products requires different strategies. The macromolecular structure of low DS high molar mass starches has been investigated by Size-Exclusion Chromatography (SEC) [6,7,8] and Asymmetrical Flow Field Flow Fractionation (AF4) coupled with Multiangle Laser Light Scattering (MALLS) [9,10,11]. The localization of acetyl groups was approached by liquid-state Nuclear Magnetic Resonance (NMR) [12] and by combining specific enzymatic hydrolyses and Matrix Assisted Laser Desorption Ionization-Time of Flight (MALDI-TOF) Mass Spectroscopy [13] or SEC [6,14,15,16]. Yet, when the DS increases, water solubility becomes lower, as a result the macromolecular analyses of highly modified starch become very difficult as no adequate investigation systems can be run in adapted organic solvents. In addition to the poor solubility, the low accessibility to the enzymes caused by the chemical groups grafted onto the starch macromolecules prevents the use of specific enzymes [17]. Because of these difficulties, so far, only very few molecular characterizations performed on highly acetylated starch (DS > 2) with high molar mass amylopectins have been reported in literature except with liquid-state NMR and SEC-MALLS [18,19], this latter being inappropriate for amylopectin analysis [20,21].

To design new chemically modified starch-based products, a better understanding of their structure-properties relationships is mandatory. Thereby, accessing structural knowledge at different scales, and in particular at the molecular scale, is essential. In order to answer the characterization needs of starch-based materials developers, the aim of this work was then to develop new tools combination to investigate the molecular structure, localization of substituents and supramolecular structure of high molar mass hydrophobically modified starches with DS from 0.8 up to 3 using original analytical approaches.

The molecular structure was investigated by combining new sample preparation strategies with chromatographic techniques including AF4-MALLS, solid and liquid-state NMR, Raman and Time of Flight Secondary Ion Mass (TOF-SIMS) spectroscopies and Atomic Force Microscopy (AFM). The first step consisted in selecting and implementing the various techniques then, the analytical strategy was developed to characterize the supramolecular organization and the localization of substituents for two types of modified starches: highly acetylated starches and etherified starches with various substituents (Figure 1). The amorphous structuration was revealed by solid-state NMR which also emerged as the best way to obtain the DS for acetylated starches. Molar mass decrease and specific ways of chain breakage due to modification were highlighted and correlated with the amylose content, while changes in supramolecular structure were associated with the DS.

## 2. Results and Discussion

### 2.1. Development of the Experimental Strategy

In order to develop a new general strategy for the multiscale structural characterization of chemically modified starches insoluble in water and with high molar mass, we focused first on one acetylated sample obtained from waxy maize starch (WMS), i.e., without amylose.

#### 2.1.1. Determination of the Degree of Substitution

The degree of substitution (DS) of the acetylated waxy maize starch (AWMS) (average number of acetyls grafted per anhydroglucose unit, AGU) was determined by three different methods: titration as previously described [22], liquid-state ^1^H-NMR in dimethylsulfoxide-d_6_ (DMSO-d_6_) and solid-state ^13^C-NMR. The DS values obtained in liquid-state (titration and ^1^H-NMR) were 1.8 ± 0.2 whereas solid-state ^13^C-NMR allowed to obtain the expected value of 2.6 (± 0.01). This indicated a bias for liquid-state methods which could result from the poor solubility of AWMS. This limitation of the liquid-state NMR had already been reported for acetylated starches with DS greater than 2 [23]. In fact, with uncomplete solubilization, the detected signals did not represent the entire sample, hence a characterization of the soluble fraction only and therefore a partial view of the material. Solid-state NMR seemed then to be the most adapted method for the DS determination of acetylated starches with DS > 2.

#### 2.1.2. Macromolecular and Molecular Structure

First AWMS solubility was evaluated. It was soluble in pure dimethylsulfoxide (DMSO) but insoluble in water (Appendix A) as well as in DMSO/water mixtures. High Performance Size-Exclusion Chromatography (HPSEC) analysis of high molar mass non-modified starches [20] had shown to be inappropriate in DMSO and we confirmed that for AWMS (results not shown). In addition, perform AF4 analysis, a more suitable technique for high molar mass starches, was not possible in DMSO (even with a system compatible with organic solvents). For these reasons, AWMS was analyzed after a deacetylation process in order to increase its water solubility and allow its analysis by aqueous AF4-MALLS. The deacetylation process was developed specifically for AWMS in order to avoid starch chains depolymerization by KOH (see the Materials and Methods section), and the complete deacetylation was checked by Fourier Transformed Infrared Spectroscopy FTIR (Appendix A). The solubilization recoveries (in water for native and in 0.1 M KOH for acetylated starch) were good (> 93%) and elution recoveries were 93% for native starch and lower for AWMS (75%) which meant that the analysis was representative. The lower elution recovery obtained for AWMS was probably due to the loss of few small sugars produced by the acetylation process through the membrane during the fractionation procedure.

The AF4 elugram of WMS exhibited one peak at 17–21 mL (Figure 2a), corresponding to amylopectin as expected [21,24]. The elugram of AWMS showed in addition a peak shoulder at 17 mL which corresponded to the elution of smaller macromolecules (as elution volume is proportional to molecular size), produced by a breakage of amylopectin during the acetylation process. Molar mass distributions and the weight average molar mass ${\bar{M}}_{w}$) also showed slightly lower values for AWMS (Figure 2a, Table 1), which confirmed the slight amylopectin degradation. Yet the dispersity (${\bar{M}}_{w}$/${\bar{M}}_{n}$) and the *z*-average radius of gyration ($\bar{R}$_G_) were stable for WMS after acetylation (Figure 2b), in line with small molar mass variations.

Debranching enzymes do not work with more than 20% DMSO in the solution and acetyl groups inhibit their action. In order to determine the chain length distribution of AWMS, it was first deacetylated to allow its solubilization in aqueous solvent and to allow the enzymes to act on it reliably. The distributions of debranched chains for deacetylated AWMS and WMS displayed a major peak corresponding to a degree of polymerization (DP) 11–13 with a large tail (Figure 2b and Appendix A), as expected for waxy maize starch [25]. Larger proportions (37.5%) of short A chains (DP 6–12) in the amylopectin cluster [1] and smaller proportions (55.9%) of intermediate chains (DP 13-36, defined as B1a and B1b chains in amylopectin [1]) were observed for AWMS compared to WMS which showed 28.4% and 65.6% for the short A and intermediate B chains respectively (Figure 2b and Appendix A). This meant that even if the molar mass of WMS was only slightly affected by the acetylation process, specific chain breakings occurred and the amylopectin fine structure was modified. B1a and B1b chains in the clusters (DP 13-36 [1]), i.e., chains with intermediate length that bear at least one chain, seemed to be cut more specifically and short A chains liberated (DP 6-12).

#### 2.1.3. Composition, localization of Acetyl Groups and Supramolecular Structure

The composition and supramolecular structure of AWMS and its native counterpart were studied by solid-state ^13^C-NMR on powdered samples. Spectral chemical shifts of native WMS were in accordance with literature [26,27]. AWMS spectrum was characteristic of acetylated species, with signals at 20.2 and 170.6 ppm assigned to the acetyl carbons (methyl and carbonyl respectively) (Figure 3). AWMS exhibited a C-6 shift from 61 ppm (for WMS, results not shown) to 65 ppm due to acetylation on C-6, and a C-1 shift from 100 ppm to 96 ppm due to acetylation on C-2, as was previously assigned by liquid-state ^13^C-NMR in D_2_O [12]. A slight shift from 73 to 70 ppm was also observed for the C-3 signal which signed for acetylation of this carbon as well, even if it was in slighter proportions. This shift included probably the C-2 shift. Indeed, the poor resolution of this spectral region did not allow us to clearly discriminate the signal of C-2 from that of C-3.

Moreover, the decomposition of C-1 signal (90–105 ppm) with Lorentz-Gauss function was performed to identify the local short-range organization of starches [28]. Native WMS showed three peaks for C-1 (results not shown), standing for A-type double helices, in line with the A-type crystallinity known for native maize starch [27]. For AWMS, the decomposition of C-1 peak (90–105 ppm) gave three resonances that have already been identified on amorphous starches [27,28,29] (Figure 3): 95.4 ppm (73% of total signal) typical for amorphous very unfavorable and/or constrained conformations for α-1,4 and α-1,6 linkages, 99.7 ppm (26% of total signal) typical for B-type double helices or paracrystalline bundles in amorphous samples (isolated double helices or double helices embryos, on a scale small enough to not give any long-range ordering visible with wide-angle X-Ray scattering) and 91.6 ppm (2% of total signal) corresponding to reducing extremities.

C-1 and C-6 signals showed typical amorphous starch behavior, as expected for such gelatinized starches and in agreement with the absence of crystallinity observed by X-Ray diffraction [30], with a minor proportion of residual double helices and a major proportion of constrained conformations for α-1,4 and α-1,6 linkages that could be attributed to the acetylation points on C6 but also on C2 and C3.

#### 2.1.4. Surface Characterization of Materials

The surface characterization was made by Raman spectroscopy on powder and film (Figure 4a). The most suitable analysis conditions were determined on native powdered starch and allowed to attribute the chemical shifts of the starch backbone linkages. In native WMS, intense bands relative to organized amylopectin (CCC around 480 cm^−1^) as well as glycoside ring vibrations (COC) were detected [31]. In AWMS the introduction of acetyl groups was visible with the vibration around 1740 cm^−1^ attributed to the deformation of the C=O bond [18,32]. The disappearance of CCC vibration around 480 cm^−1^ confirmed the loss of crystalline structure of WMS with acetylation process as was evidenced by solid-state ^13^C-NMR. Nevertheless, Raman spectroscopy did not allow characterizing the repartition of hydrophobic groups nor the DS.

The surface composition of AWMS film was approached at the µm scale by TOF-SIMS. Identification of ionized fragments from substituents followed by 2D mapping revealed a uniform distribution of acetyl fragments on film surface on the 500 µm^2^ observed area (Figure 4b and Appendix A). No remaining granules were observed which was in agreement with the absence of Maltese cross under polarized light (results not shown) and confirm that all starch granules have been destroyed during the acetylation process. AFM observations, which did not reveal any particular topography (Figure 4c), confirmed the homogeneity and smoothness of AWMS film surface.

### 2.2. Characterization of Acetylated Starches with Various DS and Amylose Content

Acetylated starches with DS of 2.4 and 0.7 (± 0.01) determined by solid-state ^13^C-NMR, and obtained from starches with amylose contents of 22 and 35% (potato and pea starches, respectively) were studied by comparison of AWMS (0% amylose) and their native counterparts using the strategy established in the previous section.

#### 2.2.1. Macromolecular Characteristics

Macromolecular characteristics obtained by aqueous AF4-MALLS for acetylated potato starch (APOS) and acetylated pea starch (APES) after deacetylation, and their native counterparts are reported in Figure 5 and Table 2. Solubilization recoveries were higher than 89% for all the starches whereas elution recoveries were 40 and 85% for APOS and APES respectively (Table 2) which indicated a higher loss of small sugars produced by the acetylation process during the analysis procedure for APOS.

AF4 elugrams exhibited two peaks for native potato (POS) and pea (PES) starches: a main peak corresponding to amylopectin (17–21 mL), and a minor peak (13–15 mL) corresponding to amylose (Figure 5), in agreement with previously reported distributions for native potato and maize starches [21,24]. Acetylated starches elugrams showed a peak at 12–17 mL which signed the presence of smaller macromolecules resulting from the scission of starch macromolecules during the acetylation process. This phenomenon was accompanied by a shift of the amylopectin population to lower macromolecular sizes, which indicated amylopectin degradation. Molar mass distributions also showed lower values for acetylated starches (Figure 5), which confirmed the amylopectin breakage during the acetylation process. This macromolecular degradation was particularly important for APOS (Figure 5a), for which the elugram did not show any amylopectin peak.

Molar mass decrease of 75% and 85% were observed for PES and POS respectively (Table 2) whereas only 13% of the molar mass was loss during the acetylation of WMS (Table 1). The dispersities of PES and POS were higher than WMS as expected because of the presence of amylose. Moreover, when it was stable between WMS and AWMS, it increased a lot for PES after acetylation (from 3.49 to 5.71) because of the high production of short macromolecular chains. This meant that amylopectin but also amylose molecules were degraded during the acetylation process. On the contrary, the dispersity of POS decreased because of a dramatic degradation of the largest macromolecules in APOS. The low elution recovery of APOS was in line with the high amount of small macromolecules in this sample: a large part of these molecules were too small compared to the cut off of the AF4 membrane and were not analyzed because they were eliminated during the fractionation process. This result was also confirmed on non-deacetylated APOS by the higher proportion of reducing units observed by solid-state ^13^C-NMR for this sample (see next section). Starch macromolecules breakage have been reported to depend on acetylation process [18], thus the low molar mass and macromolecular size of APOS is probably due to the particularly drastic process applied to this starch to obtain a DS of 2.4.

#### 2.2.2. Chain-Length Distributions of Acetylated Starches

The distributions of debranched chains of APOS and APES after deacetylation and corresponding native starches displayed two peaks (Figure 6a,b and Appendix A): a major peak corresponding to DP 11–13 and a minor peak corresponding to DP 44–49. Such distributions were expected for native starches [25,33,34,35,36]. These chain length (CL) distributions showed similar patterns with some differences depending on the amylose content and the DS. APES and APOS had more chains of DP >37 (>13.3%) than AWMS (6.5%) because of the presence of amylose, whereas AWMS and APOS, which had a DS ≥ 2.4, had more very short A chains (Figure 2b, Figure 6a,b and Appendix A).

APOS had larger proportions (75.0%) of short A chains (DP 6–12) and intermediate B1a chains (DP 13–24) and lower proportions (14.4%) of long chains (DP > 37) compared to POS which showed 69.3% and 18.3% of DP 6-24 chains and DP >37 chains respectively (Figure 6c and Appendix A). This decrease of long chains was in line with specific chain breakings on the long B2 and B3 chains of potato amylopectin [1] during the acetylation process and agreed well with the drastic molar mass decrease observed by AF4-MALLS that corresponded to the liberation of low molar mass dextrins. The same chain breaking trend was observed for PES, exhibiting the highest amylose content, yet in very slight proportions (Figure 6b and Appendix A) compared to POS. This was due to a less drastic acetylation process in link with its lower DS (0.7), by analogy to the observations made on acetylated barley starches by Bello-Pérez et al. [19] which reported a decrease of long chains and an increase of short chains with an increasing effect when the DS increase. Moreover these slight variations of CL distribution of amylopectin despite a decrease of 75% of the molar mass of PES caused by the acetylation process could also be related to a preferential hydrolysis of amylose chains and extra-long amylopectin chains in starches containing amylose. Whereas for starches without amylose, such as WMS, a preferential hydrolysis of chains with intermediate length was observed in the amylopectin during the acetylation process (Figure 6c and Appendix A).

#### 2.2.3. Localization of Acetyl Groups, Supramolecular Structure and Surface Characteristics of Acetylated Starches

Solid-state ^13^C-NMR spectra obtained on powdered samples showed the typical acetyl carbons signals at 20.2–20.4 and 169.4–170.60 ppm for both APOS and APES, and exhibited a C-6 shift from 61 ppm for the native starches (results not shown) to 65 ppm due to acetylation (Figure 7). Only APOS and AWMS spectra showed a shift for the C-1 resonance from 100 ppm to 96 ppm (Figure 2a,b) indicating they were acetylated on C-2, in line with the results presented in the previous section, and a shift of C-3 resonance (from 72 ppm to 70 ppm).

Thus, AWMS and APOS were also acetylated in C-2 and C-3 whereas APES, which had a lower DS, was acetylated only in C-6. This privileged substitution on C-6 had already been identified for acetylated rice starches with DS < 1 [22] and was in agreement with the reactivity of hydroxyl groups: the primary hydroxyl group on C6 was the most reactive and accessible, followed by the secondary hydroxyl groups on C2 and C3 [37].

The decomposition of C-1 signal allowed the identification of local short-range organization of starches. PES showed three peaks for C-1, like WMS, standing for A-type double helices, and POS two peaks (results not shown), indicative of B-type double helices, as expected [27,29]. For APOS the decomposition of C-1 peak gave the same three resonances typical for amorphous starches found for AWMS but with different proportions (Figure 6a): 64% of total signal corresponded to typical for amorphous material very unfavorable and/or constrained conformations linked to the acetylation points on C6, C2 and C3 (95.4 ppm), 33% of total signal to typical for B-type double helices (99.6 ppm) and 3% of total signal to reducing extremities (91.6 ppm). This confirmed the absence of crystallinity in APOS like in AWMS. Moreover the higher amount of amorphous material very unfavorable and/or constrained conformations found in AWMS and the lower B-type double helices could be explained by the absence of amylose in AWMS, and by the high amount of very short chains found in AWMS (as chains with DP < 10 could not be involved in double helices), respectively. Besides APES showed the most defined C-1 signal which was deconvoluted onto five main resonances (Figure 7b): 102.5 ppm typical for V-type single helices in amorphous starches and branching points for amylopectin (41%), 99.6 ppm corresponding to B-type double helices or paracrystallinity (24%), 96.9 ppm typical for constrained α-1,6 linkages (19%), 94.2 ppm typical for constrained α-1,4 linkages (15%) and 91.2 ppm corresponding to reducing extremities (1%). APES had then a higher local order than the two other acetylated starches which could be related to its lower macromolecular degradation (Figure 6; Figure 7 and also attested by a lower amount of reducing extremities) and a lower acetylation. Increasing the DS would then produce not only the destruction of crystallinity but of local order in amorphous starch as well. Though C-1 signals showed higher local organization, C-6 signal for APES was typical of amorphous starch, indicating that the substitution was preferentially on amorphous regions, near branching points, confirming the hypothesis made by Shogren [38] for low substituted corn starch.

The surface characterization of APOS and APES was made by Raman spectroscopy (Figure 7c). The introduction of acetyl groups was visible (1740 cm^−1^) and the disappearance of CCC vibration (~480 cm^−1^) observed for APOS, as well as for AWMS, but not for APES, which allowed to follow the crystalline structure loss when DS increased. APES spectrum only contained C-O-C bonds and had a band around 1260 cm^−1^ attributed to V-helices [31]. Raman spectroscopy confirmed on the surface area the supramolecular information obtained in bulk by solid-state ^13^C-NMR and X-ray diffraction (results not shown). Moreover, Raman spectroscopy imaging showed a homogeneous repartition of acetyl groups on the surface whatever the DS (results not shown).

The surface composition of starch films at the µm scale, approached by TOF-SIMS, revealed a uniform distribution of acetyl groups for APOS and APES (Appendix A) and confirmed that all starch granules have been destroyed during the acetylation process independently of the DS and the amylose content. Under AFM, APES film surface showed large nodules (approximately 2 μm, Figure 7d) and surface roughness, consistent with the observations made by Hong et al. [39] on cassava acetylated starches with low DS. These authors showed that higher roughness can prove a good compatibility despite a heterogeneous distribution of hydrophobic groups on starch matrix. On the opposite, AWMS film surface did not reveal any particular topography (Figure 4c). According to the previous hypothesis on low-acetylated starch surface, this could indicate a more reliable compatibility when higher substitution in spite of a probable covering of starch matrix by hydrophobic groups.

### 2.3. Characterization of Etherified Starches

The four starch ethers were studied according to the analytical strategy used for starch acetates.

#### 2.3.1. Degree of Substitution

The DS of starch ethers were determined by liquid (liq) ^1^H-NMR (DS_liqNMR_), for the modified starches soluble in NMR solvents and for the others by elemental analysis (EA) (Table 3).

A DS_HDo.liqNMR_ of 0.40 (average number of 2-hydroxydodecyl grafts per AGU) was determined for HDo-POS-1 by ^1^H-NMR analysis of the product dissolved in DMSO-d6 using the formulas reported in Appendix A. This DS of 0.40 was confirmed for HDo-POS-1 by EA, using the formulas demonstrated in Appendix A from %C (DS_(HDo.EA(C)) = 0.39) and from %H (DS_(HDo.EA(H)) = 0.41). A DS of 1.6 was determined for HDo-POS-2 (insoluble in NMR solvents) by EA with the same formulas (DS = 1.60 from %C and 1.58 from %H).

By analyzing the ^1^H-NMR spectra of the two mixed starch ethers in CDCl_3_ and in THF-d8, the DS corresponding to 2-hydroxydodecyl/2-hydroxyphenethyl groups (calculated for the two types of grafts as demonstrated in Appendix A) were found to be equal to 1.45/0.10 for HDo-HPhe-POS-1, and to 1.67/0.19 for HDo-HPhe-POS-2.

#### 2.3.2. Macromolecular Characteristics

Etherified starches were insoluble in water and soluble or partially soluble in DMSO and THF (Appendix A). Because of these poor solubilities, we were able to analyze the macromolecular characteristics of HDo-POS-1 only without removing the grafted groups. After a first solubilization in pure DMSO, the DMSO content in the solution was reduced to 5% with water addition (final concentration of HDo-POS-1: 0.5 mg mL^−1^) in order to be able to carry on the macromolecular characteristics analysis of HDo-POS-1 by aqueous AF4-MALLS with good elution recovery (91%). HDo-POS-1 exhibited a minor peak at 20 mL and a major peak at 14 mL (Figure 8). This latter corresponded to a large amount of small macromolecules, produced by the hydrolysis of the amylopectin chains during the etherification process. On the other hand, the residual fraction of amylopectin in HDo-POS-1 showed a higher molar mass distribution (Figure 8) and a higher $\bar{R}$_G_ (241 nm) than the POS amylopectin (Figure 8, $\bar{R}$_G_ of 179 nm), in line with its later elution (20 mL instead of 19 mL for POS amylopectin) which signed for a higher hydrodynamic radius. Thus, even if the chemical modification induced differences in the dn/dc which could considerably affect the results obtained with MALLS, the increase of size of the amylopectin fraction after chemical modification was obvious. Moreover, the residual amylopectin fraction seemed to be densified, as for the same elution volume, i.e., the same macromolecular size, it exhibited a higher molar mass. This increase of size and densification could not be solely due to an increase of molar mass caused by grafting on the amylopectin chains; more likely it could be explained by a supramolecular structuration of the prototypes in the aqueous medium thanks to hydrophobic interactions between the newly grafted chains. The prototype macromolecules could indeed aggregate by forming denser structures with a hydrophobic core with the fatty chains and a hydrophilic ring corresponding to the amylopectin chains.

#### 2.3.3. Composition, Localization of Ether Groups and Supramolecular Structure

Figure 9a shows the solid-state ^13^C-NMR spectra of starch ethers. Even if some typical signals for fatty chain (12.5–32.5 ppm) were observed for both HDo and HDo-HPhe grafted starches, the signal at 127.5 ppm corresponding to the phenyl was only visible for HDo-HPhe-POS-1 (DS 1.45/0.10), which did not allow determining the DS from these spectra. Moreover, contrary to ester starches with carbonyl function that influenced carbon shifts depending on the location, ether did not have characteristic shifts thus it was not possible to localize functionality.

As the signal resolution of starch on solid-state ^13^C-NMR spectrum was low, we could not deconvolute the spectra and detect the characteristic resonances of semi-crystalline or amorphous starch, except for HDo-POS-1, which was less substituted. C-1 signal deconvolution of HDo-POS-1 showed the major presence of V-type single helices (102.5 ppm) which could be the result of the complexation of the linear chains of D-glucosyl units linked in α-1,4 with the 2-hydroxydodecyl grafted groups. In HDo-POS-1, C-1 and C-6 signals showed typical amorphous starch pattern, as well as C-4 signal and C-2, C-3, C-5 peak [27] which signed for crystalline structure loss during chemical modification as was also observed for acetylated starches. This was in agreement with the absence of crystallinity observed by X-Ray diffraction for all these etherified starches [40].

On Raman spectra (Figure 9b), we identified characteristic vibrations from aliphatic chains (1080–1125, 1300, 1450 cm^−1^) and supplementary aromatic linkages for HDo-HPhe-POS-1 and HDo-HPhe-POS-2 (999, 1025 and 1601 cm^−1^). HDo-POS-1 only exhibited the organized-polysaccharide C-C-C vibration at 480 cm^−1^, this was consistent with well-resolved starch signals in solid-state ^13^C-NMR and signed for a higher structuration of the amorphous starch in this sample. In addition, a typical vibration at 1260 cm^−1^ corresponding to V-type helices was detected in both HDo-POS samples, with a lower signal in HDo-POS-2, but was absent in HDo-HPhe-POS samples. The amorphous structuration of chemically modified samples at the surface would then decrease with DS by analogy with acetylated samples, and probably decrease as well with the size of the grafted group.

Films surface composition determined by TOF-SIMS (Figure 9c and Appendix A) for HDo-POS-1 and HDo-POS-2 exhibited uniform distribution of polysaccharide and alkyl fragments without phase separation indicating that its repartition did not depend on composition and reagents ratio. TOF-SIMS spectra and 2D mapping of HDo-HPhe-POS-1 and HDo-HPhe-POS-2 showed also uniformly distributed fragments from alkyl chains (upon tetramer) (Figure 9c and Appendix A) and phenyl derivatives even in low intensity, probably because of a low substitution on surface (Appendix A). The smaller and more intense zones of about 20 μm observed in HDo-HPhe-POS-1 maybe caused by a difference in the height of the surface.

## 3. Materials and Methods

### 3.1. Materials

Native waxy maize (WMS), potato (POS) and pea (PES) starches containing 0, 22 and 35% amylose respectively were obtained from Roquette Frères (Lestrem, France). Three corresponding acetylated starches, named AWMS, APOS and APES respectively, were produced by Roquette Frères in homogeneous media, as previously described by Quettier [41]. APOS was obtained with a basic catalyst whereas the others were produced with an acid catalyst. Isoamylase (EC 3.2.1.68) was from Sigma-Aldrich (St. Louis, MO, USA). Four starch ethers were obtained by reaction of POS with epoxides in basic aqueous media by classical protocols [42,43,44]. Among them, two 2-hydroxydodecyl POS (HDo-POS-1 and 2) were synthesized by reaction of POS with different amounts of 1,2-epoxydodecane (1,2-EDD), and the two others are mixed ethers of 2-hydroxydodecyl 2-hydroxyphenethyl POS named HDo-HPhe-POS-1 and 2 which were obtained by reactions of POS with two different mixtures of 1,2-EDD and styrene oxide (SO) (Figure 1). The structural and physical characterizations of materials were made on 200 µm thick films elaborated from raw powder by compression molding at 200 °C. For pea acetate starch, 40% wet basis of water was added to the raw powder prior to compression molding.

### 3.2. Methods

The degree of substitution (DS) of modified starches were determined for acetylated starches by titration as previously described [22] and NMR, and for etherified starches by elemental analysis (Appendix A) and NMR.

#### 3.2.1. Nuclear Magnetic Resonance Measurements

The DS were determined by liquid-state ^1^H-NMR using a 300 MHz or 400 MHz spectrometer (Bruker, Wissembourg, France) after solubilization of the samples in deuterated dimethylsulfoxide (DMSO-d_6_) at 80 °C for acetylated starches as previously described [18] and for etherified starches (Appendix A); after solubilization in tetrahydrofuran (THF-d_8_, ambient temperature) or in chloroform (CDCl_3_, ambient temperature) (Appendix A). Solid-state ^13^C-NMR experiments were carried on a Bruker Avance III 400 WB NMR spectrometer operating at 100.62 MHz for ^13^C (B_0_ = 9.4 T), equipped with a double-resonance H/X CP-MAS 4-mm probe for Cross-Polarization Magic Angle Spinning (CP-MAS) solid-state experiments. The samples were packed in 4 mm ZrO2 rotors. The magic-angle-spinning (MAS) rate was fixed at 9 kHz and each acquisition was recorded at room temperature (293 °K). Chemical shifts were calibrated with external glycin, assigning the carbonyl at 176.03 ppm. The chemical shift, half-width and area of peaks were determined using a least-squares fitting method using the Peakfit^®^ software (Systat Software Inc., San Jose, CA, USA). The DS of acetylated starches were determined with solid-state ^13^C-NMR based on the methods developed in liquid-state ^1^H-NMR [18].

#### 3.2.2. Determination of size, Molar Mass and Chain Length Distributions

Pretreatment of the samples

Acetylated starches (10 mg) were first deacetylated in KOH 1 M (1 mL) for 1 day at room temperature under mild stirring. This procedure also allowed their solubilization without degradation of the molar mass of starch polysaccharides. Actually, this procedure was carefully adjusted in order to avoid macromolecular degradation of the chains by β-elimination mechanism due to the pH of this solvent. In particular, stirring, temperature (4 °C and 25 °C) and time of solubilization (1 h, 3 h, 24 h, 2 days, 5 days and 7 days) were tested (results not shown). Samples solutions were then diluted 10 times with water. Native starches were first solubilized in DMSO, precipitated with ethanol, dried and then solubilized in water by microwave heating under pressure, as previously described [24]. Etherified starches were first solubilized in DMSO at 10 mg mL^−1^, and then diluted 20 times with water, as they were soluble in DMSO/water mixture. All the starches solutions were filtered on 5 µm Durapore^TM^ membranes (Waters, Bedford, MA, USA) before analysis.

Determination of macromolecular characteristics

Molar mass and radius of gyration distributions were determined by AF4-MALLS. The AF4 equipment was an Eclipse system (Wyatt Technology Corporation (WTC), Santa Barbara, CA, USA) with a AF4 long channel (tip–to–tip length of 291 mm) placed in a ThermosPRO oven regulated at 25 °C and an ISO-3000SD pump (Thermo Scientific, Waltham, MA, USA). A 350 µm polyester spacer and a regenerated cellulose membrane with a cutoff of 10 KDa (Merck Millipore, Darmstadt, Germany) were used. The carrier (0.02% NaN_3_ in Millipore water) was filtered through a 0.1 µm Durapore membrane (Millipore), and degassed. All samples were introduced into the channel using an autosampler WPS-3000SL (Thermo Scientific) and eluted with a flow method adapted from Rolland-Sabaté et al. [21]. The cross flow (F_c_) was set at 0.84 mL min^−1^ for the sample introduction (injection at 0.20 mL min^−1^ for 300 s) and the focusing/relaxation period (300 s). F_out_ was then set at 0.84 mL min^−1^ and F_c_ decreased in 480 s from 0.4 to 0.05 mL min^−1^, maintained 600 s at 0.05 mL min^−1^, and finally 300 s at 0 mL min^−1^. Two on-line detectors were used: a MALLS instrument (Dawn^®^ HELEOS™, WTC, Santa Barbara, CA, USA) fitted with a K5 flow cell and a GaAs laser (λ = 658 nm), and an refractometer (Optilab, WTC, Santa Barbara, CA, USA) operating at the same wavelength (WTC). Solubilization and elution recovery rates, and acquisition and data processing using ASTRA^®^ software from WTC were established as previously described [21,24].

Determination of chain length (CL) distributions

Deacetylated starches solubilized at 1 mg mL^−1^ in 0.1 M KOH were neutralized with 0.1 M HCl and debranched using isoamylase (591 U g^−1^ of dry starch) at 40 °C for 24 h in 50 mM acetate buffer pH 3.6 with 0.02% NaN_3_ and 0.1% BSA. The CL distributions of debranched deacetylated starches were examined by high-performance anion-exchange chromatography coupled with pulsed amperometric detection (HPAEC-PAD) (ICS-500+, ThermoFisher), using a Carbopac PA200 (4 mm × 250 mm) column. The elution process involved 500 mM NaOH (eluent A), 1 M NaOAc (eluent B) and water eluted at 0.4 mL min^−1^. The elution gradient was composed of 30% eluent A and (i) 0–30 min with a linear gradient from10% to 25% eluent B and 60% to 45% water, (ii) 30–52.5 min with a second linear gradient from 25% to 34% eluent B and 45% to 36% water, (iii) 52.5–82.5 min with a third linear gradient from 34% to 40% eluent B and 36% to 30% water, (iv) 82.5–85 min with a fourth linear gradient from 40% to 50% eluent B and 30% to 20% water. The concentration of each chain was determined by using the linear relationship between the detector response per mole of α(1,4) chains and CL. The linear curve coefficients were determined from maltooligosaccharide standards of DP1 to 7 and were used for longer compound quantification. It should be noticed that as in HPAEC-PAD response coefficients decrease with increasing DP, the % area of long chains is not representative of the exact weight fraction of each DP.

#### 3.2.3. Topology and Surface Chemical Composition

Raman spectroscopy analyzes were carried out on powders and films by a LabRAM Visible Raman micro-spectrometer (Horiba Jobin-Yvon, Kyoto, Japan), with a He-Ne laser (λ = 632.8 nm) and a network of 600 t mm^−1^. The device was equipped with a confocal microscope (Olympus, Tokyo, Japan) with a ×100 magnification lens. Spectral resolution was 1.1 cm^−1^ per pixel. The data were acquired and processed with NGSLabSpec (version 5.45.09).

Time of flight secondary ion mass spectroscopy (TOF-SIMS) spectra were obtained on films using a TOF SIMS5 (IONTOF GmbH, Münster, Germany) comprising a pulsed ion source (Bi^3+^) with a current of 0.35 pA and with charge compensation. Both positive and negative secondary ion spectra were collected for each sample with a range of *m*/*z* = 0–200 Da and accumulated from 50 scans. Three spectra were recorded for each sample on 3 different areas of 500 µm × 500 µm with 128 × 128 pixels each.

Atomic force microscopy (AFM) analyzes were carried out on films by a NTEGRA atomic force microscope (NT-MDT, Moscow, Russia), under air, with a semi-contact mode (lever and tetrahedral silicone tip 14 to 16 μm high and doped with antimony). The resonance frequency was 320 kHz.

## 4. Conclusions

Macromolecular and molecular characterizations of highly substituted starches by conventional methods are a real challenge because of their low solubility in aqueous solvents and their resistance to enzyme action. We developed new analytical strategies (Figure 10) allowing, for the first time, the determination of macromolecular parameters, chemical composition and topography for highly acetylated and etherified starches with various substituents showing the almost universal character of this analytical approach which can thus help the design of eco-friendly materials.

In particular, despite limitations linked to the solubility of the samples, we showed that the NMR technique, used in its liquid and solid aspects, allowed obtaining the DS of low to highly modified starches and localizing grafted groups. Moreover, using solid-state ^13^C-NMR we showed that the loss of crystallinity as well as the modification of the amorphous structure depended on the process, the DS and the nature of the grafted groups. Finally, thanks to AF4-MALLS, which permitted the size and molar mass distributions analysis of starches with high molar mass, we showed for the first time that the chemical modification process caused reduction of molar mass depending on the process, the DS, the nature of the grafted groups and the amylose content for high molar mass starches with DS from 0.4 to 3.

## Figures and Tables

**Figure 1 molecules-25-02827-f001:**
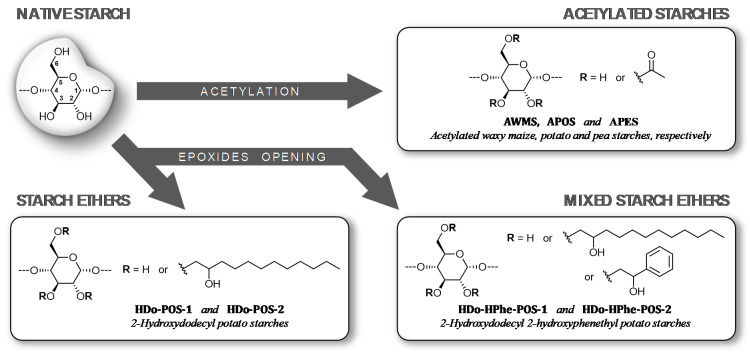
Synthesis, representation and codification of modified starches.

**Figure 2 molecules-25-02827-f002:**
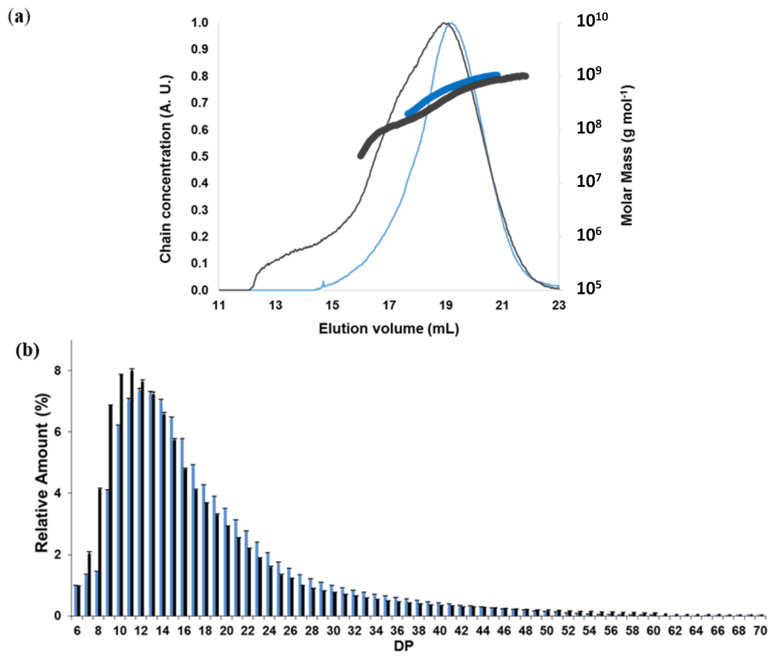
Macromolecular structure of native and acetylated waxy maize starch. (**a**) Elugrams (thin lines) and molar mass distributions (thick lines) of native WMS (blue) and acetylated WMS (AWMS) after deacetylation (black) obtained by AF4-MALLS; (**b**) Chain length distributions of WMS (Blue) and AWMS after deacetylation (Black). DS before deacetylation: 2.6.

**Figure 3 molecules-25-02827-f003:**
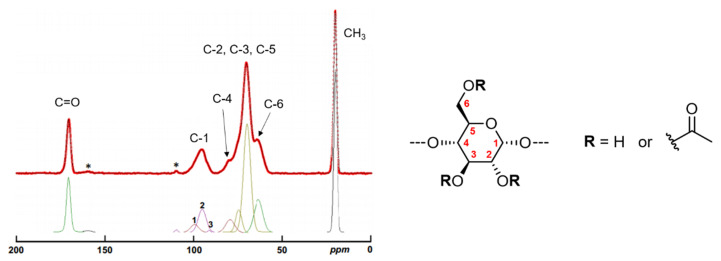
Solid-state ^13^C-NMR spectrum obtained for acetylated waxy maize starch (DS 2.6) and its spectral decomposition. * Spinning side band.

**Figure 4 molecules-25-02827-f004:**
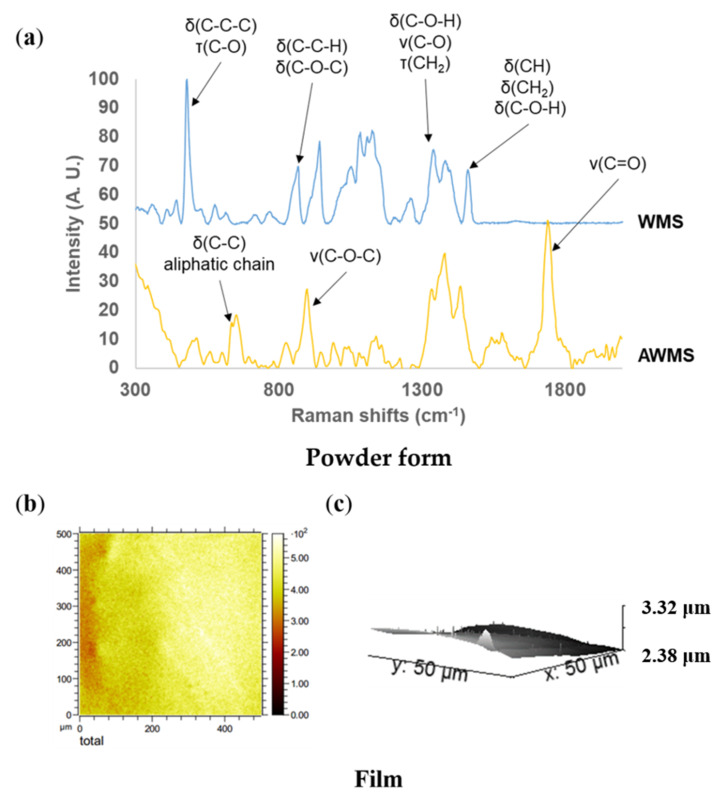
Surface characterization of acetylated waxy maize starch (DS 2.6). (**a**) Raman spectra of WMS and AWMS, (**b**) TOF-SIMS 2D mapping of the acetate fragment in AWMS, (**c**) AFM topography image of AWMS.

**Figure 5 molecules-25-02827-f005:**
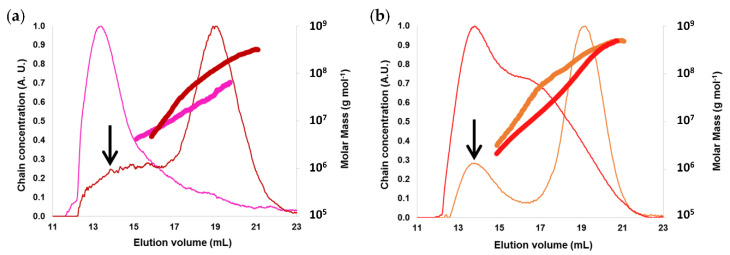
Macromolecular characteristics of native and acetylated potato and pea starches obtained by AF4-MALLS. (**a**) POS (dark red) and APOS after deacetylation (pink); (**b**) PES (orange) and APES after deacetylation (red). Elugram (thin lines): chain concentration versus elution volume; molar mass distributions (thick lines); the arrows indicate the amylose peak. Native potato (POS), pea (PES) starches and corresponding acetylated starches: APOS and APES, respectively. The DS of APOS and APES before deacetylation were 2.4 and 0.7, respectively. ${\bar{M}}_{w}$, $\bar{R}$_G_ and ${\bar{M}}_{w}$/${\bar{M}}_{n}$.

**Figure 6 molecules-25-02827-f006:**
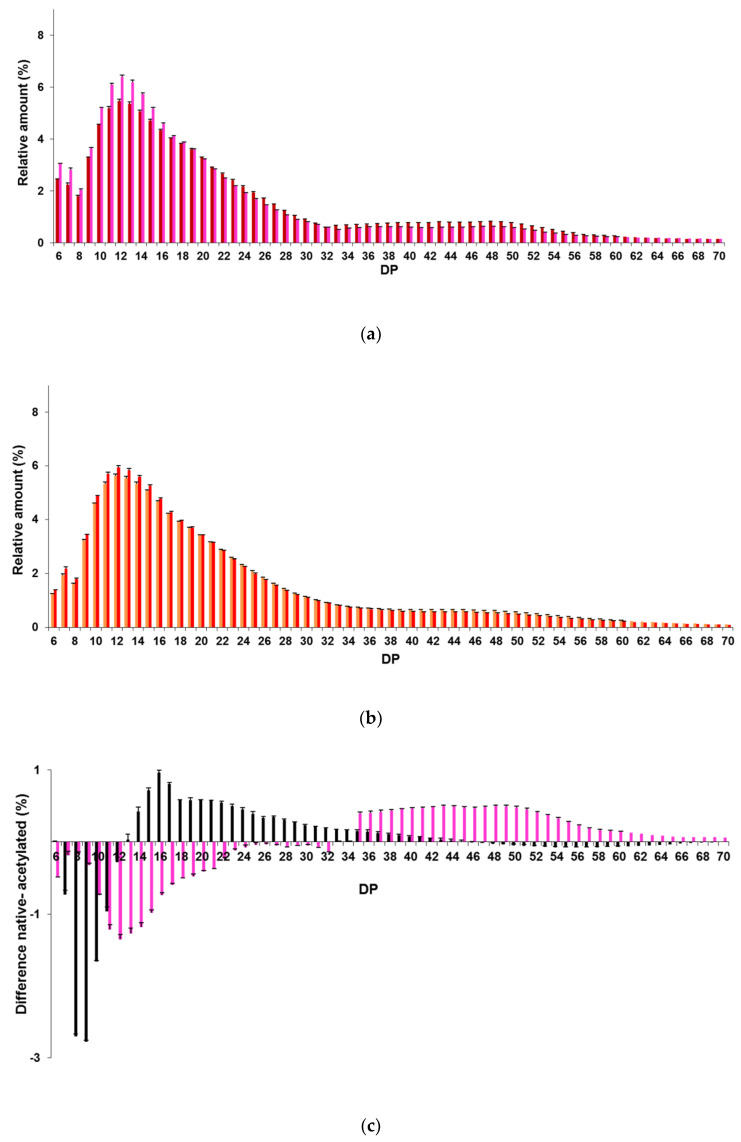
Chain length distributions of native and acetylated starches. (**a**) POS (dark red) and APOS (pink), (**b**) PES (Orange) and APES (Red), (**c**) difference of chain length between native and acetylated starches for AWMS (Black) and APOS (pink). Native waxy maize (WMS), potato (POS), pea (PES) starches and corresponding acetylated starches: AWMS, APOS and APES, respectively. The DS of AWMS, APOS and APES before deacetylation were 2.6, 2.4 and 0.7, respectively.

**Figure 7 molecules-25-02827-f007:**
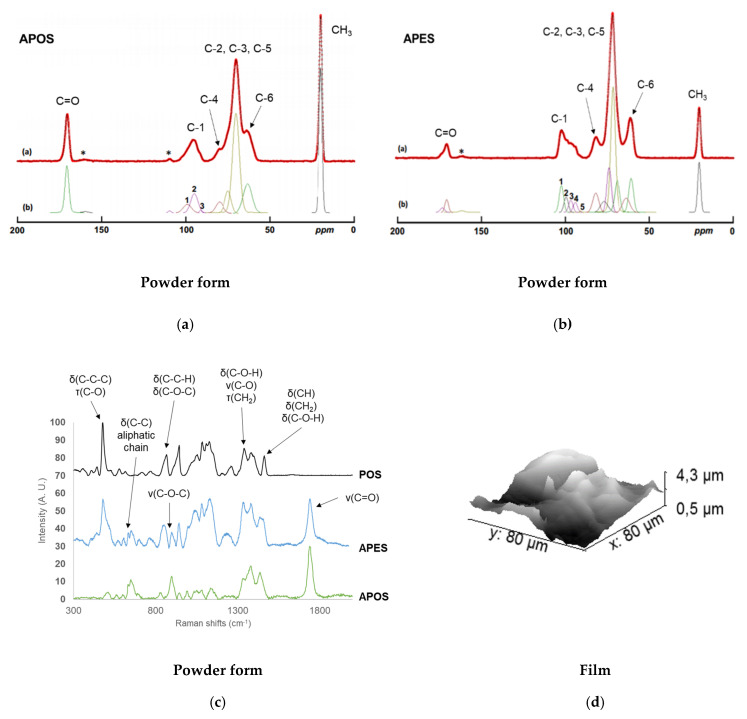
Bulk and surface characterization of APES and APOS Solid-state ^13^C-NMR spectra obtained for APOS (**a**) and APES (**b**) and their spectral decomposition, Raman spectra of POS, APOS and APES (**c**), AFM topography image of APES (**d**). The DS of APOS and APES before deacetylation were 2.4 and 0.7, respectively. *Spinning side band.

**Figure 8 molecules-25-02827-f008:**
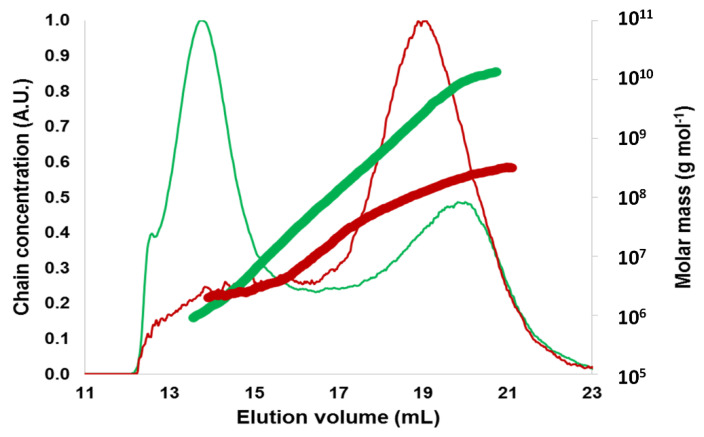
Macromolecular characteristics of native and etherified potato starches obtained by AF4-MALLS. Elugrams, i.e., chain concentration versus elution volume (thin lines), and molar mass distributions (thick lines) for POS (dark red), and HDo-POS-1 (green). Native potato (POS) starch and corresponding etherified starch with DS 0.40: HDo-POS-1.

**Figure 9 molecules-25-02827-f009:**
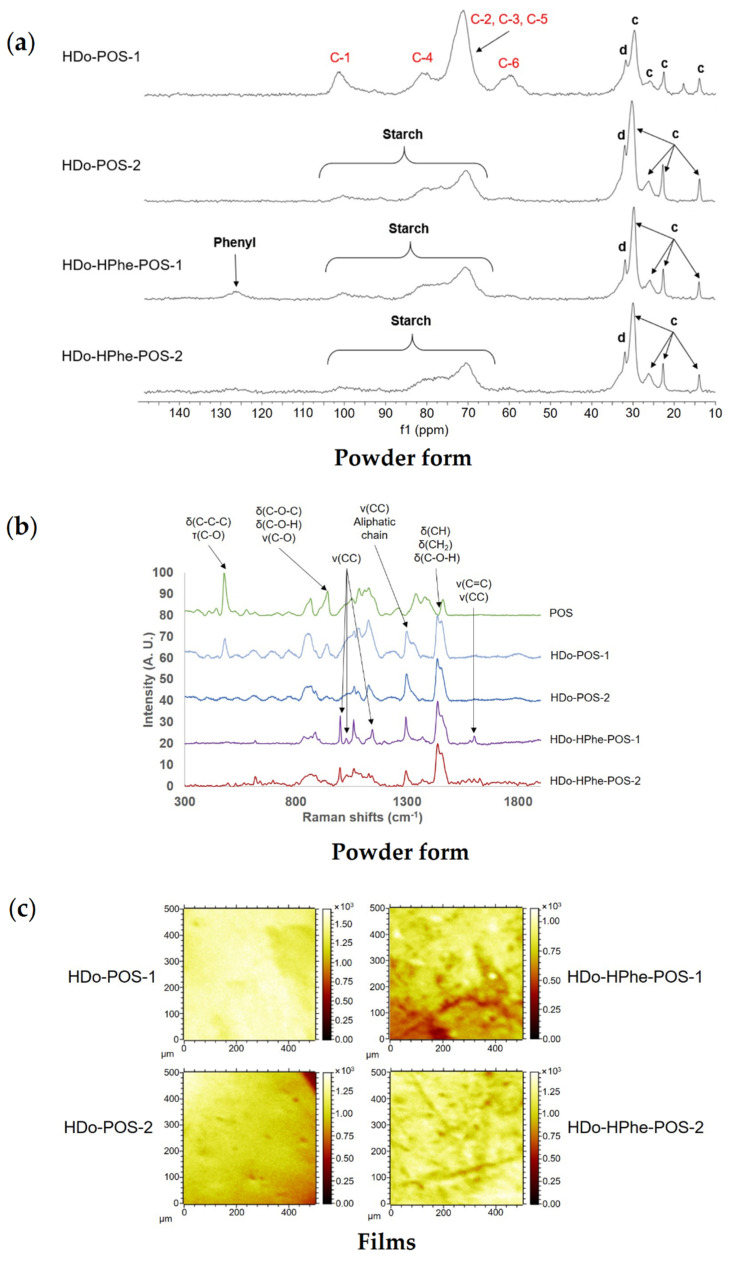
Bulk and surface characterization of ethers produced from potato starch. (**a**) Solid-state ^13^C-NMR spectra obtained for HDo-POS-1, HDo-POS-2, HDo-HPhe-POS-1 and HDo-HPhe-POS-2 and their spectral decomposition, (**b**) Raman spectra of potato starch (POS) and corresponding ether starches (HDo-POS-1, HDo-POS-2, HDo-HPhe-POS-1 and HDo-HPhe-POS-2), (**c**) TOF-SIMS 2D mapping of POS ethers films. The DS of HDo-POS-1, HDo-POS-2, HDo-HPhe-POS-1 and HDo-HPhe-POS-2 were 0.40, 1.60, 1.45/0.10, and 1.67/0.19, respectively.

**Figure 10 molecules-25-02827-f010:**
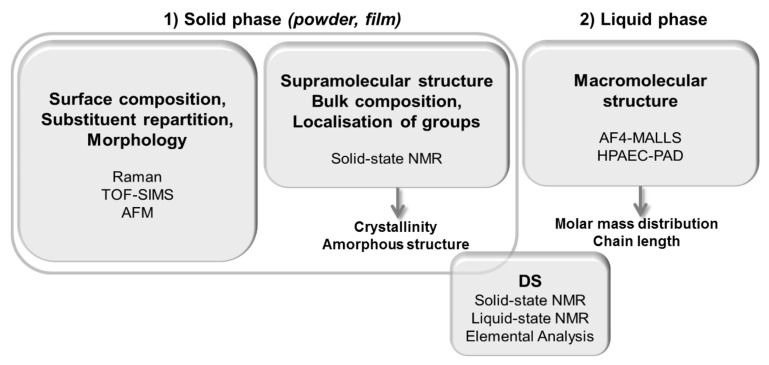
Designed strategy for multi-scale and multi-state characterization of high molar mass chemically-modified starches with 0.4 < DS ≤ 3.

**Table 1 molecules-25-02827-t001:** Weight-average molar mass (${\bar{M}}_{w}$), *z*-average radius of gyration ($\bar{R}$_G_) and dispersity index (${\bar{M}}_{w}$/${\bar{M}}_{n}$) determined by AF4-MALLS for WMS and AWMS after deacetylation.

Reference	${\bar{M}}_{w}\times 10^{-7}\;(g\;{\mathrm{mol}}^{-1})$	${\bar{R}}_{G}\;(nm)$	${\bar{M}}_{w}$ $/{\bar{M}}_{n}$
WMS	60.0	282	1.12
AWMS	52.0	283	1.16

${\bar{M}}_{w}$,
$\bar{R}$_G_ and ${\bar{M}}_{w}$ /${\bar{M}}_{n}$ values were taken over the whole peak. The experimental uncertainties were 5%.

**Table 2 molecules-25-02827-t002:** Solubilization and elution recoveries, weight-average molar mass (${\bar{M}}_{w}$), *z*-average radius of gyration ($\bar{R}$_G_) and dispersity index (${\bar{M}}_{w}$/${\bar{M}}_{n}$) determined by integrating the whole elugrams.

Reference	Solubilization Recovery (%)	Elution Recovery (%)	${\bar{M}}_{w}\times 10^{-7}\;(g\;{\mathrm{mol}}^{-1})$	${\bar{R}}_{G}\;(\mathrm{nm})$	${\bar{M}}_{w}$ $/{\bar{M}}_{n}$
POS	100	70	10.0	179	4.79
PES	89	100	20.4	219	3.49
APOS	100	40	1.5	104	1.35
APES	100	85	5.1	181	5.71

${\bar{M}}_{w}$, $\bar{R}$_G_ and ${\bar{M}}_{w}$/${\bar{M}}_{n}$ values were taken over the whole peak. The experimental uncertainties were 5%.

**Table 3 molecules-25-02827-t003:** Degrees of substitution of starch ethers determined by different techniques.

Starch Ethers	DS_HDo.liqNMR_ ^a^ (DMSO-d_6_)	DS_HDo.EA(C)_ ^b^	DS_HDo.EA(H)_ ^c^	DS_HDo.liqNMR_ ^d^ (CDCl_3_ and THF-d_8_)	DS_HPhe.liqNMR_ ^e^ (CDCl_3_ and THF-d_8_)
HDo-POS-1	0.40 ± 0.05	0.39 ± 0.05	0.41 ± 0.06	ND ^f^	ND ^f^
HDo-POS-2	ND ^g^	1.60 ± 0.09	1.58 ± 0.11	ND ^f^	ND ^f^
HDo-HPhe-POS-1	ND ^g^	-	-	1.45 ± 0.07	0.10 ± 0.02
HDo-HPhe-POS-2	ND ^g^	-	-	1.67 ± 0.04	0.19 ± 0.03

^a^ Number of 2-hydroxydodecyl groups (HDo) per AGU determined by ^1^H-NMR in DMSO-d6, ^b^ by elemental analysis from wt% of C, ^c^ by elemental analysis from wt% of H, ^d^ by ^1^H-NMR in CDCl_3_ and THF-d8. ^e^ Number of 2-hydroxyphenethyl groups (HPhe) per AGU determined by ^1^H-NMR in CDCl_3_ and THF-d8. ^f^ Products insoluble in chloroform and in tetrahydrofuran. ^g^ Products insoluble in dimethylsulfoxide. ND: Not determined.

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
