# Peer review of "Multiscale Structure of Starches Grafted with Hydrophobic Groups: A New Analytical Strategy"

_molecules, 2020, doi:10.3390/molecules25122827_

Round 1

Reviewer 1 Report

Regardless its biological origin starch performs poorly when used as bioplastic. Physical, chemical, physicochemical and enzymatic modifications of starch provide materials performing better in this respect. Authors of this paper focused on a series of acylated starches and starch ethers as possible bioplastics of better functional properties. Acylation and etherification of starches is known for several decades. Such kinds of modification are quite promising in this respect although there are known several more efficient modifications. Because interest in bioplastics spreads widely I indicated a moderate interest in the content of reviewed paper limited solely to acylated and etherated starches.

The quality and performance of bioplastics made of starches modified by acylation and etherification depends somehow also on applied reaction conditions. Linking performance of those bioplastics with the structure of resulting products and selected reaction conditions would be beneficial. It could lead to tailoring bioplastics of required functional properties by selecting proper starch variety, modification method and reaction conditions. Unfortunately, thus far, it is impossible because there are problems with recognizing structure of modified starches. These problems are associated with limited possibilities of commonly available analytical technics.

In presented study authors extended number of instrumental techniques (Asymmetric Flow Field Fractionation associated with Multiangle Laser Light Scattering) for recognizing structural details of a series of acylated starches and starch ethers. That approach provides an convincing evidence that such strategy can be suitable for tailoring properties of bioplastics.   

In spite of some false interpretations of details (see below) elaborated strategy can be extended to other modifications of polysaccharides and even lower saccharides. This paper needs a minor revision.

Going into details, authors should introduce the following improvements:

-First sentence of Introduction (line 44) is incorrect. Certainly the World resources of cellulose and hemicelluloses exceed resources of starch.

-Fig. 3.  (lines 157-159): starred side bands are improperly considered rotational bands. At rotation of 9 kHz these side bands should be located on such distance from the main band. In fact they are located on the distance of 1462 and 1196 Hz and, additionally, these distances are not equal to one another.

-Interpretation of 13C NMR spectrum: The bands distributed into peaks (according to Gauss or Lorentz?) are called “doublet” and “triplet”. In fact they are singlets grouped around a certain value and they do not result from any spin-spin coupling.

-Line 278: a change in the position of C2 on which substitution occurs could be mentioned.

-Considerable mass increase on etheration with epoxides not necessarily results only from polymerization of epoxide. It can also be an effect of the phase contrast (differences in the refraction indices of a molecule and liquid phase). This effect can considerably affect the results taken with MALLS. 

As non-native English speaker I may not evaluate the quality of the language

Author Response

Regardless its biological origin starch performs poorly when used as bioplastic. Physical, chemical, physicochemical and enzymatic modifications of starch provide materials performing better in this respect. Authors of this paper focused on a series of acylated starches and starch ethers as possible bioplastics of better functional properties. Acylation and etherification of starches is known for several decades. Such kinds of modification are quite promising in this respect although there are known several more efficient modifications. Because interest in bioplastics spreads widely I indicated a moderate interest in the content of reviewed paper limited solely to acylated and etherated starches.

This paper reports the development of a global strategy for the analysis of chemically modified starches. This strategy would be employed not only on acetylated and etherified starches but also to characterize the structure of other kind of chemically modified starches even if some adaptations should be implemented depending on the modifications.

The quality and performance of bioplastics made of starches modified by acylation and etherification depends somehow also on applied reaction conditions. Linking performance of those bioplastics with the structure of resulting products and selected reaction conditions would be beneficial. It could lead to tailoring bioplastics of required functional properties by selecting proper starch variety, modification method and reaction conditions. Unfortunately, thus far, it is impossible because there are problems with recognizing structure of modified starches. These problems are associated with limited possibilities of commonly available analytical technics.

In presented study authors extended number of instrumental techniques (Asymmetric Flow Field Fractionation associated with Multiangle Laser Light Scattering) for recognizing structural details of a series of acylated starches and starch ethers. That approach provides an convincing evidence that such strategy can be suitable for tailoring properties of bioplastics.   

In spite of some false interpretations of details (see below) elaborated strategy can be extended to other modifications of polysaccharides and even lower saccharides. This paper needs a minor revision.

Going into details, authors should introduce the following improvements:

-First sentence of Introduction (line 44) is incorrect. Certainly the World resources of cellulose and hemicelluloses exceed resources of starch.

The sentence was modified as follows: “Starch is, after cellulose and hemicelluloses, one of the most abundant carbohydrates in plants”

-Fig. 3.  (lines 157-159): starred side bands are improperly considered rotational bands. At rotation of 9 kHz these side bands should be located on such distance from the main band. In fact they are located on the distance of 1462 and 1196 Hz and, additionally, these distances are not equal to one another.

This remark is correct, but the reviewer did not associate the spinning side band (SSB) with the correct peak. Indeed, the SSB located at 160.4 ppm is to be associated with the peak at 70.5 ppm, and the rotation band at 110 ppm with the peak at 20.2 ppm, which corresponds to differences of 9009 and 8986 Hz respectively. The equal SSB is present in the acquisition file (data not shown).

-Interpretation of 13C NMR spectrum: The bands distributed into peaks (according to Gauss or Lorentz?) are called “doublet” and “triplet”. In fact they are singlets grouped around a certain value and they do not result from any spin-spin coupling.

The authors remove the mention of doublet and triplets and replace it by “peaks”. They also modified the materials and methods section as follows: “Solid state 13C-NMR experiments were carried on Bruker Avance III 400 WB NMR spectrometer operating at 100.62 MHz for 13C (B0 = 9.4 T), equipped with a double-resonance H/X CP-MAS 4-mm probe for CP-MAS (Cross-Polarization Magic Angle Spinning) solid-state experiments. The samples were packed in 4 mm ZrO2 rotors. The magic-angle-spinning (MAS) rate was fixed at 9 kHz and each acquisition was recorded at room temperature (293 °K). Chemical shifts were calibrated with external glycin, assigning the carbonyl at 176.03 ppm. The chemical shift, half-width and area of peaks were determined using a least-squares fitting method using the Peakfit® software (Systat Software Inc., USA).”

The authors added a mention of the deconvolution process in the text: “Moreover, the decomposition of C-1 signal (90–105 ppm) with Lorentz-Gauss function was performed to identify the local short-range organization of starches [28].”

-Line 278: a change in the position of C2 on which substitution occurs could be mentioned.

A C-1 shift from 100 ppm to 96 ppm had been previously identified to be due to acetylation on C-2, by liquid-state 13C-NMR in D2O [12].  The authors modified the text as follows L278: “Only APOS and AWMS spectra showed a shift for the C-1 resonance from 100 ppm to 96 ppm (Figures 2a and b) indicating they were acetylated on C-2, in line with the results presented in the previous section, and a shift of C-3 resonance (from 72 ppm to 70 ppm). ”

The authors also added a mention in the text L157: “AWMS exhibited a C-6 shift from 61 ppm (for WMS, results not shown) to 65 ppm due to acetylation on C-6, and a C-1 shift from 100 ppm to 96 ppm due to acetylation on C-2, as was previously assigned by liquid-state 13C-NMR in D2O [12]. A slight shift from 73 to 70 ppm was also observed for the C-3 signal which signed for acetylation of this carbon as well, even if it was in slighter proportions. This shift included probably the C-2 shift. Indeed, the poor resolution of this spectral region did not allow us to clearly discriminate the signal of C-2 from that of C-3.”

-Considerable mass increase on etheration with epoxides not necessarily results only from polymerization of epoxide. It can also be an effect of the phase contrast (differences in the refraction indices of a molecule and liquid phase). This effect can considerably affect the results taken with MALLS. 

The authors agreed with the reviewer, MALLS results could be biased by the modification of the dn/dc values due to grafting of fatty chains. Nevertheless, the increase of size was also observed thanks to AF4 fractionation as a later elution in AF4 signs for a bigger object. In consequence, the authors modified the text as follows: “On the other hand, the residual fraction of amylopectin in HDo-POS-1 showed a higher molar mass distribution (Figure 8) and a higher G (241 nm) than the POS amylopectin (Figure 8, G of 179 nm), in line with its later elution (20 mL instead of 19 mL for POS amylopectin) which signed for a higher hydrodynamic radius. Thus, even if the chemical modification induced differences in the dn/dc which could considerably affect the results obtained with MALLS, the increase of size of the amylopectin fraction after chemical modification was obvious. Moreover, the residual amylopectin fraction seemed to be densified, as for the same elution volume, i.e., the same macromolecular size, it exhibited a higher molar mass. This increase of size and densification could not be solely due to an increase of molar mass caused by grafting on the amylopectin chains; more likely it could be explained by a supramolecular structuration of the prototypes in the aqueous medium thanks to hydrophobic interactions between the newly grafted chains.”

As non-native English speaker I may not evaluate the quality of the language

Reviewer 2 Report

The manuscript reports the evaluation and demonstration of new analytical technique combinations to analyze the degree of substitution, molecular weight, and chain breakage of acetylated starches, which have potential applications as bioplastics.   The authors present a well-written, comprehensive analysis of acetylation using various techniques and provide thorough discussion of the results.  Figures are designed well, clear, and include descriptive captions.  As such, it is recommended that this manuscript be accepted for publication.

Author Response

The authors have read carefully the manuscript and checked it for the spelling and English language, and for the references list.

Reviewer 3 Report

The manuscript under review aims to answer the characterization needs of starch-based materials developers by developing new strategies to investigate the molecular structure, localization of substituents, and supramolecular structure of high molar mass hydrophobically modified starches with DS from 0.8 up to 3 using already known analytical techniques.

Generally, the text is well written, concise, and scientifically sound. There are however few minor concerns:

  1. Supplementary info contains only the figures from the main text.
  2. Please check if all the abbreviations are explained in the text. E.g. HPSEC is not explained.
  3. On page 3, section 1.1. Determination of the Degree of Substitution

The paragraph refers to strategies employed in the literature for the determination of DS. Which was the basis to decide that solid-state NMR is the most fitted technique for DS measurement? The literature study or some experiments similar to those from the literature, performed by the authors? If experiments were performed than they should be presented at this point. If this section is not based on experimental results, perhaps it is better to move it to the introduction section.

Author Response

The manuscript under review aims to answer the characterization needs of starch-based materials developers by developing new strategies to investigate the molecular structure, localization of substituents, and supramolecular structure of high molar mass hydrophobically modified starches with DS from 0.8 up to 3 using already known analytical techniques.

Generally, the text is well written, concise, and scientifically sound. There are however few minor concerns:

  1. Supplementary info contains only the figures from the main text.

The Supplementary materials contain 7 sections with different tables and figures and different from the main text. Thus the authors do not understand this remark.

  1. Please check if all the abbreviations are explained in the text. E.g. HPSEC is not explained.

The authors checked the abbreviations and added the corresponding explanations in the text and in the supplementary materials.

  1. On page 3, section 1.1. Determination of the Degree of Substitution

The paragraph refers to strategies employed in the literature for the determination of DS. Which was the basis to decide that solid-state NMR is the most fitted technique for DS measurement? The literature study or some experiments similar to those from the literature, performed by the authors? If experiments were performed than they should be presented at this point. If this section is not based on experimental results, perhaps it is better to move it to the introduction section.

This section is based on experiments. The authors modified the text as follows:

“The degree of substitution (DS) of the acetylated waxy maize starch (AWMS) (average number of acetyl grafted per anhydroglucose unit, AGU) was determined by three different methods: titration as previously described [22], liquid-state 1H-NMR in dimethylsulfoxide-d6 (DMSO-d6) and solid-state 13C-NMR. The DS values obtained in liquid-state (titration and 1H-NMR) were 1.8 ± 0.2 whereas solid-state 13C-NMR allowed to obtain the expected value of 2.6 (± 0.01). This indicated a bias for liquid-state methods which could result from the poor solubility of AWMS. This limitation of the liquid-state NMR had already been reported for acetylated starches with DS greater than 2 [23]. In fact, with uncomplete solubilization, the detected signals did not represent the entire sample, hence a characterization of the soluble fraction only and therefore a partial view of the material. Solid-state NMR seemed then to be the most adapted method for the DS determination of acetylated starches with DS > 2.”

And the authors modified the methods section as follows:

“3.2. Methods

The degree of substitution (DS) of modified starches were determined for acetylated starches by titration as previously described [22] and NMR, and for etherified starches by elemental analysis (Supplementary Materials S6) and NMR.

3.2.1. Nuclear Magnetic Resonance Measurements

The DS were determined by liquid-state 1H-NMR using a Bruker 300 MHz or 400 MHz spectrometer after solubilization of the samples in dimethylsulfoxide (DMSO-d6) at 80 °C for acetylated starches as previously described [18] and for etherified starches (Supplementary Materials S6); after solubilization in tetrahydrofuran (THF-d8, ambient temperature) or in chloroform (CDCl3, ambient temperature) (Supplementary Materials S7).”